# The Abundance of Tumor-Infiltrating CD8^+^ Tissue Resident Memory T Lymphocytes Correlates with Patient Survival in Glioblastoma

**DOI:** 10.3390/biomedicines10102454

**Published:** 2022-10-01

**Authors:** Marco Pio La Manna, Diana Di Liberto, Marianna Lo Pizzo, Leila Mohammadnezhad, Mojtaba Shekarkar Azgomi, Vincenzo Salamone, Valeria Cancila, Davide Vacca, Costanza Dieli, Rosario Maugeri, Lara Brunasso, Domenico Gerardo Iacopino, Francesco Dieli, Nadia Caccamo

**Affiliations:** 1Central Laboratory of Advanced Diagnosis and Biomedical Research (CLADIBIOR), University of Palermo, 90129 Palermo, Italy; 2Department of Biomedicine Neurosciences and Advanced Diagnostics, School of Medicine, University of Palermo, 90127 Palermo, Italy; 3Department of Health Promotion, Mother and Child Care, Internal Medicine and Medical Specialties, University of Palermo, 90129 Palermo, Italy; 4Tumor Immunology Unit, Department of Health Sciences, University of Palermo, 90127 Palermo, Italy; 5Neurosurgical Clinic, AOUP “Paolo Giaccone”, Post Graduate Residency Program in Neurologic Surgery, 90127 Palermo, Italy

**Keywords:** glioblastoma, tissue resident memory cells, CD8^+^ lymphocytes, tumor microenvironment

## Abstract

Glial tumors alone account for 40% of all CNS tumors and present a low survival rate. The tumor microenvironment is a critical regulator of tumor progression and therapeutic effectiveness in glioma. Growing evidence from numerous studies of human solid tumor-infiltrating CD8^+^ T cells indicates that tissue-resident memory T cells (TRM) represent a substantial subpopulation of tumor-infiltrating lymphocytes (TILs). Although it is reported that some types of cancer patients with high immune infiltration tend to have better outcomes than patients with low immune infiltration, it seems this does not happen in gliomas. This study aimed to characterize TRMs cells in the glioma tumor microenvironment to identify their potential predictive and prognostic role and the possible therapeutic applications. Fluorescence activated cell sorting (FACS) analysis and immunofluorescence staining highlighted a statistically significant increase in CD8^+^ TRM cells (CD103^+^ and CD69^+^ CD8^+^ T cells) in gliomas compared to control samples (meningioma). In-silico analysis of a dataset of *n* = 153 stage IV glioma patients confirmed our data. Moreover, the gene expression analysis showed an increase in the expression of TRM-related genes in tumor tissues compared to normal tissues. This analysis also highlighted the positive correlation between genes associated with CD8^+^ TRM and TILs, indicating that CD8^+^ TRMs cells are present among the infiltrating T cells. Finally, high expression of Integrin subunit alpha E (ITGAE), the gene coding for the integrin CD103, and high CD8^+^ TILs abundance were associated with more prolonged survival, whereas high ITGAE expression but low CD8^+^ TILs abundance were associated with lower survival.

## 1. Introduction

Gliomas define a remarkably heterogeneous group of cancers of the central nervous system originating from the glia cells. In 2016, a new WHO classification system was introduced based on the integration between morphological aspects and tumor molecular alterations, thus providing phenotypic and genotypic parameters that provide diagnostic process objectivity [1,2]. This classification system includes five main categories of adult diffuse glioma: glioblastoma, IDH-wild type; glioblastoma, IDH-mutant; diffuse or anaplastic astrocytoma, IDH-wild type; Diffuse or anaplastic astrocytoma, IDH-mutant, and oligodendroglioma or anaplastic oligodendroglioma, IDH-mutant, and 1p19q co-deletion. In addition, based on the histological aspect in the absence of molecular determinants, a sixth group is identified as ‘malignant glioma NOS’ (not otherwise specified). Globally, the incidence of diffuse adult gliomas differs substantially between countries [2,3,4]. In adults over forty, the age-adjusted annual incidence of astrocytic tumors is 6.8/per 100,000 people. North European countries have a higher incidence rate than countries with a predominantly Asian or African population [5]. Glioblastoma (GBM), the most common primary brain tumor, exhibits a particularly infiltrating character and highly aggressive clinical behavior. Survival is still low today, the prognosis is poor, and the predictive classification is complex, despite the more integrated classification, the aggressive standards of treatment regimens, and the most recent novelties on the therapeutic front [6,7,8]. Furthermore, while immunotherapy has been highly successful in various types of cancer, no survival advantage has been observed in patients with GBM [8]. Therefore, it is crucial to identify new prognostic biomarkers and therapeutic targets.

Over the past decade, numerous studies have shown that the tumor microenvironment (TME) is a critical regulator of tumor progression and therapeutic efficacy in GBM [9]. The TME of GBM is unique in its cellular composition and accessibility to immune cells since it includes cells with powerful immunoregulatory properties without a typical lymphatic drainage system [8,9,10]. It has been shown that in GBM, the number of infiltrating myeloid-derived suppressor cells (MDSC) capable of inhibiting the functions of T cells and NK cells increases, giving rise to a highly immunocompromised TME [9,10,11]. Furthermore, cytokines such as TGF-β promote the recruitment and survival of regulatory T cells, associated with a worse prognosis in many types of cancer. It is also known that the infiltration of immune cells into TME reflects the state of the immune system and can predict the prognosis of cancer [9]. 

TILs represent an essential component of the microenvironment of brain tumors. These cells could be critically involved in tumor growth, progression, and/or control. TILs represent a small percentage of all cells in both GBM (approximately 2.5%) and other CNS tumors and usually consist of T cells and, to a lesser extent, NK and B lymphocytes [10]. Growing evidence indicates that a substantial subpopulation of TILs is represented by TRMs [12]. This subpopulation is emerging as a tumor-specific subset of T cells since their presence appears to characterize the microenvironment of various human cancers, intuitively implying their role in the antitumor immune response and possible prognostic and therapeutic applications [12,13,14].

CD8^+^ TILs and intraepithelial T lymphocytes express two integrins, CD103 and LFA-1, whose expression is regulated by TGF-β, particularly abundant in the tumor microenvironment and whose immunosuppressive effect is exploited by tumor cells to escape the immune response. CD103, together with the activation marker CD69 and the integrin CD49a, defines the CD8^+^ TRM T lymphocyte subset. TGF-β plays a crucial role in the formation and maintenance of these cells, inducing the expression of the ITGAE gene, coding for the CD103 molecule [12]. Data from various studies revealed that intratumorally CD103^+^ TRM cells frequently express PD-1, TIM-3, and LAG-3, which are involved in the induction of functional exhaustion and their dysfunction at the tumor site [12,15]. It seems that TGF-β is involved in the induction of PD-1 in CD8^+^ T cells, contributing to their anergy and tolerance [12]. Regarding the functional activity of intratumoral TRM cells, they express *IFNG* (IFN-γ), Granzyme A, (*GZMA*), and Granzyme B (*GZMB*) [12,16].

Further functional studies have shown that CD8^+^ CD103^+^ TILs cells can secrete inflammatory cytokines, including IFN-γ and TNF-α [12,16]; this phenomenon is optimized by the interaction of CD103 with the E-cadherin of the target tumor cells. The expression of *GZMB* seems to distinguish between CD8^+^ TRM T lymphocytes with a cytotoxic function and those with a regulatory role in autoimmune and tumor pathologies [12].

Furthermore, the unique properties of TRM cells, their potential spectrum of action, and their predictive value have aroused a growing interest in these cells for their possible use as a new therapeutic target in cancer therapy [17].

Although the positive prognostic role of CD103^+^ TRM cells has been widely demonstrated in multiple types of cancer [13,14,16], it remains to be clarified and investigated what role this cellular subset plays in gliomas. In this study, we assessed the consistency of TRM T lymphocytes infiltrating GBM, compared to low-grade meningiomas, used as a control. We also analyzed GBM datasets to highlight the differential expression of TRM-related genes.

## 2. Materials and Methods

### 2.1. Patients

Patients were recruited between September 2020 and June 2021 at the U.O.C. of Neurosurgery of the AOUP “Paolo Giaccone” of Palermo and the Department of Neurosurgery of the Hospital “Ospedali Riuniti Villa Sofia-Cervello” of Palermo. For our study, 20 patients (10 females and 10 males) between the ages of 48 and 83 years (median age of 63 years) underwent surgical resection and were definitively diagnosed with GBM (100% of the sample with histological diagnosis of GBM IV degree). The diagnosis of GBM was made under 2016 WHO CNS tumor grading and staging system. Nine biopsy samples from patients diagnosed with meningioma (6 females and 3 males) between the ages of 34 and 77 years (median age of 57 years) were used as the control group. All of the recruited patients signed informed consent and the study was approved by the AOUP Paolo Giaccone Ethical committee Palermo 1, approval code: 8, approval date 14 September 2022.

### 2.2. Biopsies Digestion

The biopsy samples were subjected to initial mechanical and subsequent enzymatic digestion under sterile conditions. Mechanical digestion was performed by placing the biopsy sample on a Petri dish and obtaining small fragments from it using a scalpel. The fragments thus obtained were collected in RPMI medium (Euroclone) supplemented with HEPES (Euroclone) 20 mM and antibiotics (100 U/mL of penicillin and 100 mg/mL of streptomycin) and transferred to wells of a 24-well multiwell plate. In each well, we added hyaluronidase (Sigma, Merk Life Science S.r.l. Milan italy, 20 µg/mL), DNAse (Sigma, 50 µg/mL), Type IV collagenase (Life Technologies, Carlsbad, CA, USA, 1.5 mg/mL). The biopsies, treated as previously described, were placed in an incubator at 37 °C for about 90 min. At the end of the incubation, the enzymatic digestion process was stopped by filtration on a 40 µm filter to remove the fibrous component in RPMI medium added with 10% fetal bovine serum (FCS) and 2 mM L-glutamine. The cells recovered from filtration were centrifuged at 1600 rpm for 5 min. Then, the cell pellet was resuspended in 1 mL of complete RPMI medium for cell counting.

### 2.3. Flow Cytometry

The cells obtained from the digestion of the biopsies were washed and resuspended in PBS, then collected and transferred into 5 mL tubes for flow cytometry, and then incubated for 15 min in the dark with the Zombie Violet™ fixable viability stain (Biolegend, San Diego, CA, USA). The cells were then washed and resuspended in FACS buffer (PBS, 2% FBS, 2 mM FBS EDTA) to proceed with the labeling with antibodies (and their isotopic controls) for the recognition of surface markers. For immunophenotyping of T lymphocytes and TRM, the cells were labeled with anti-CD3 FITC clone UCHT1, anti-CD103 PE clone Ber-ACT8, anti-CD69 PerCP-Cy5.5 clone FN50 (Biolegend, San Diego, CA, USA), anti-CD8 PE-Cy7 clone HIT8a, anti-CD45 APC-Cy7 clone 2D1. In addition, the following antibodies were used for immunophenotyping NK and γδ T cells: anti-CD56 PE clone B159, anti-CD16 PECy7 clone 3G8, anti-Vδ2 APC clone B6 (Biolegend, San Diego, CA, USA). All the antibodies used, except for anti-CD69 and anti-Vδ2, were obtained from the BD Biosciences Company, (Franklin Lakes, NJ, USA). The samples were acquired using a BD FACSAria flow cytometer at least 50,000 viable cells were acquired for each sample. The gating strategy for the detection of T lymphocytes and TRM T cells is shown in Appendix A. The data obtained were analyzed using the FlowJo software (BD Biosciences).

### 2.4. Immunofluorescence Staining

Two independent analyses were carried out in two laboratories of the University of Palermo. At the CLADIBIOR laboratory, immunofluorescence staining was performed on 5-μm-thick paraffin-embedded glioma and meningioma sections, which were used as the patients and controls, respectively. Firstly, the sections were treated to remove paraffin (de-paraffinization). Following rehydration, antigens were unmasked using Dako Target Retrieval Solution (Glostrup, Denmark; pH 9.0), according to the manufacturer’s guidelines. Primary staining was performed incubating sections with rabbit monoclonal CD8 (ab237709, Abcam, Cambridge, UK), rabbit polyclonal CD69 antibody (Cat# PA5-114989, Invitrogen, Waltham, MA, USA), and rabbit polyclonal CD103 antibody (Cat# PA5-80744, Invitrogen) diluted in phosphate-buffered saline (PBS, Sigma, St Louis, MO, USA) containing 3% bovine serum albumin and 0.05% Tween20 (PBS/BSA 0.05 TW20) overnight at 4 °C. The secondary staining was applied with goat anti-rabbit Cy5 (Cat# A10523, Invitrogen), goat anti-rabbit Alexa Fluor®555 (Cat# A-21428, Invitrogen), and goat anti-rabbit Alexa Fluor®488 (Cat# A11008, Invitrogen), diluted 1:200 in PBS/BSA 0.05 TW20 at room temperature for 1 h and 30 min. Fixation with paraformaldehyde 2% was performed between each staining step. Finally, nuclei were counterstained with Hoechst 33342 (Cat. H1399, Invitrogen) for 15 min at room temperature. Sections subjected to rehydration, and fixation stained with only secondary antibodies were used as the negative control. Lif files of images were collected by confocal laser-scanning microscope DMI6000 with Leica Application Suite X, using 40× and 63× confocal laser scanning microscopy.

#### Image Analysis

Image analyses were performed using an open-source image processing software ImageJ v1.53 [18]. Prior to performing calculations, the merged color image was separated into individual channels using the “Color split” feature, then two red and green channels, belonging to CD69^+^ and CD103^+^ cells were merged for co-expression analysis. The merged image was converted to an 8-bit image and the background was corrected through the subtract background process, which was processed using the manual threshold tool of ImageJ. In addition to these procedures, we masked the image using the “binary” process, and then the “analyze particles” tool was used to count the cell numbers. The size (pixel 2) was chosen as 20-infinity to count the cells.

At the Tumor Immunology Unit’s laboratory, four-micrometer–thick human tissue sections were deparaffinized, rehydrated, and unmasked using Novocastra Epitope Retrieval Solutions at pH9 in a thermostatic bath at 98 °C for 30 min. Subsequently, the sections were brought to room temperature, washed in PBS, and after neutralization of the endogenous peroxidase with 3% H_2_O_2_ and Fc blocking by 0.4% casein in PBS (Novocastra), sections were incubated with primary antibodies. For multiple-marker immunostaining, sections were subjected to sequential rounds of single-marker immunostaining, and the binding of the primary antibodies was revealed by the use of specific secondary antibodies conjugated with different fluorophores. The following primary antibodies were used for immunofluorescence on human tissues: rabbit anti-human CD69 (Cat# PA5-114989, Invitrogen, 1:100) and rabbit anti-human CD103 (Cat# PA5-80744, Invitrogen, ready to use).

The same sections that underwent double immunofluorescence were stained for CD4 (mouse anti-human, clone 4B12, ready to use, cod. PA0427, Leica Novocatra) or CD8 (mouse anti-human, clone 4B11, 1:20, code NCL-L-CD8-4B11, Leica Novocastra) by immunohistochemistry. Staining was revealed using SignalStain Boost IHC Detection Reagent (AP, Mouse) and Ferangi Blue was used as substrate-chromogen followed by counterstaining with Harris hematoxylin.

Slides were analyzed under a Zeiss Axioscope A1 microscope equipped with four fluorescence channels widefield IF. Microphotographs were collected using a Zeiss Axiocam 503 Color digital camera with the Zen 2.0 Software (Zeiss, Oberkochen, Germany).

### 2.5. TILs Estimation and Gene Expression Analysis

We used the public data repository, “The Cancer Genome Atlas” (TCGA), as our primary source of samples. To analyze the data generated by TCGA, we directly accessed the input data containing 153 Glioblastoma tumor, and 5 normal samples accompanied by clinical data that were obtained from TCGA (https://portal.gdc.cancer.gov/) (Data accessed 17 July 2022) and were profiled for class discovery and survival analysis. Gene expression data of GBM from TCGA were used to identify differentially expressed genes (DEGs) through the package “DESeq2” or the “linear models for microarray data” (LIMMA) methods. The DEGs were then overlapped and used for survival analysis by univariate and multivariate COX regression. Based on the gene signature of multiple survival-associated DEGs, a risk score model was established, and its prognostic and predictive role was estimated through Kaplan–Meier analysis and log-rank test. EPIC (Estimating the Proportions of Immune and Cancer cells) computational method which is an R-package available at (https://github.com/GfellerLab/EPIC) (Data accessed 17 July 2022) is used to estimate immune cells and other nonmalignant cell types found in tumors using RNA-seq-based gene expression reference profiles. Estimate R package [19] was used to evaluate tumor purity for each TCGA glioma sample (*n* = 158). This tool infers tumor purity from the expression of stromal and immune cell markers in tumor tissues. Based on reference gene expression profiles, we were able to divide the estimated infiltration percentage of TILs into two groups using the EPIC algorithm: low log_2_ transcripts per million (TPM) expression level and high TPM expression levels as we referred to low and high infiltration. 

### 2.6. Statistical Analysis

The data were analyzed using the “Multiple t-test Analysis” of GraphPad Prism version 9.0 (GraphPad, San Diego, CA, USA) and applying the Holm-Sidak method as a correction of multiple analysis. The correlation between gene expression was analyzed by the Spearman rank test. The survival between the GBM patients, differentially expressing CD8 ITGAE and CD69, was assessed by Kaplan Mayer survival analysis. The differences were significant when the *p*-value obtained was <0.05.

## 3. Results

### 3.1. TRM Cells Are Present among GBM TILs

To evaluate the composition of TILs in GBM, GBM tissues were freshly obtained from 20 GBM patients and 9 patients with low-grade meningioma as controls, and analysis of cell surface molecules defining T, NK cells, and γδ T cells was performed using flow cytometry. Cumulative data are shown in Figure 1.

Lymphocyte subsets were evaluated using cell-surface markers and indicated as a percentage of the total number of CD45 cells in each sample (Figure 1). All lymphocyte populations, apart from NK and γδ T cells, were significantly more represented in the GBM tissue than in control (meningioma). In GBM samples, CD3^+^ lymphocytes represented an average of 72.7% of the CD45^+^ population and consisted mainly of CD4^+^ and CD8^+^ T lymphocytes which accounted for 40.1% and 32.3% of viable CD45^+^ cells, respectively. Relating these values to CD3^+^ cells, CD4^+^ lymphocytes and CD8^+^ lymphocytes represented, respectively, 55.7% and 44.9% of the CD3^+^ population (Figure 1A).

The frequency of GBM-infiltrating γδ T lymphocytes and NK cells was significantly lower compared to the other cell populations and not statistically significant compared to the meningioma group. The average frequency of γδ T lymphocytes accounts for 1.7%, while NK cells accounted for 1.5% of the CD45^+^ leukocyte (Figure 1B).

CD103^+^ CD69^+^ TRM cells were also present among TILs (Figure 2) with CD8^+^ TRM cells are abundant than CD4^+^ TRM cells (9.5% vs. 3.6% of viable CD45^+^ leukocytes; *p* = 0.038). Moreover, CD8^+^ TRM cells were significantly more represented in GBM tissue than in meningioma either in their percentages (9.5% vs. 1.8%; *p* = 0.001, Figure 2A) or in their absolute counts (Figure 2B). Conversely, the percentage and absolute values of CD4^+^ TRM TILs did not statistically differ between GBM and meningioma (Figure 2A,B). 

These results show a statistically significant increase in the frequency of tumor-infiltrating CD8^+^ T cells and CD8^+^ TRM cells (CD103^+^ CD69^+^) in GBM biopsies compared to low-grade meningioma.

Since previous papers have emphasized the importance of immune cell localization within distinct tumor regions, related to the risk of tumor recurrence [20,21,22], we also visualized intratumoral TRM cells by immunofluorescence analysis on formalin-fixed, paraffin-embedded (FFPE) sections from 5 GBM (Figure 3A) and 2 meningioma samples (Figure 3B). 

In our analysis, cells positive for the CD8, CD103, and CD69 surface markers (which define the phenotype of the CD8^+^ TRM lymphocytes) were detected in the GBM sample and were counted as 15 cells in the selected area, while CD8^+^ TRM cells were absent from meningioma samples. Additional analysis from an independent laboratory that combined immunohistochemistry and immunofluorescence for CD8, CD69, and CD103 confirmed the detection of a few scattered CD69- and CD103-positive -CD8, TRM cells (Figure 3C).

### 3.2. Genes Associated with CD8^+^ TRM Cells Are Enriched in GBM and Are Associated with a Better Prognosis

We interrogated the TCGA database to determine whether *ITGAE* (CD103) and *CD69*, the two genes associated with CD8^+^ TRM T cells, were differentially expressed between GBM and meningioma patients.

To this aim, initially, we data mined an independent cohort *n* = 153 GBM transcriptomes acquired from TCGA through the Timer 2.0 software. The differential expression module analysis showed both *ITGAE* (Figure 4A) and *CD69* (Figure 4B) genes were significantly increased in GBM tissue when compared with adjacent healthy tissues.

Moreover, multiple immune deconvolution analysis of GBM-infiltrating leukocytes showed a statistically significant positive correlation between the expression of the *CD8* gene, and the genes associated with TRM T cells (Figure 5): in detail, a statistically significant positive correlation was found between the abundance of CD8 TILs and the *ITGAE* (*p* = 0.002410; Rho = 0.257). We found also a positive, but not statistically significant correlation between the abundance of CD8 TILs and CD69 (*p* = 0.092; Rho = 0.286) genes.

Overall, the results obtained by flow cytometry, immunofluorescence/immunohistochemistry, and bioinformatics show that a CD8^+^ CD103^+^ CD69^+^ TRM T lymphocyte population is significantly increased amongst GBM TILs.

Using the data available on TCGA and the EPIC software, we also analyzed the possible correlation between the expression of the *ITGAE* and *CD69* genes and molecular markers of GBM (Isocitrate Dehydrogenase 1 (*IDH1*), Isocitrate Dehydrogenase 2 (*IDH2*), Alpha-Thalassemia/mental Retardation, X-linked (*ATRX*), and Tumor Protein P53 (*TP53*)).

As shown in Figure 6A there was a statistically significant negative correlation between expression of *ITGAE* and expression of *ATRX* (Rho −0.25; *p* = 0.002) or *TP53* (Rho −0.16; *p* = 0.04), while *ITGAE* expression was not significantly correlated with *IDH-1* and *IDH-2*. 

Similarly (Figure 6B) a significantly negative correlation was found between *CD69* gene expression and *ATRX* (Rho −0.16; *p* = 0.04), *TP53* (Rho −0.20; *p* = 0.01) and *IDH-2* (Rho −0.22; *p* = 0.007), but not with *IDH-1*.

Finally, we correlated the same data set of *n* = 153 GBM patients with clinical outcomes. 

We then correlated frequencies of CD8^+^ TILs abundance and *ITGAE* and *CD69* gene expression with clinical outcomes by analyzing the independent data set mentioned previously for *n* = 153 GBM patients for whom follow-up was available. Across the whole cohort, those patients with high CD8^+^ T cell abundance and high *ITGAE* gene expression (Figure 7A, group 4) showed better cumulative survival (>80 months), as compared to those patients with a low CD8^+^ T cells abundance and high *ITGAE* gene expression (Figure 7A, group 3). Conversely, GBM patients with low CD8^+^ T cell abundance and low *ITGAE* expression (Figure 7A, group 1) had lower cumulative survival when compared to those patients with high CD8^+^ T cell abundance and low *ITGAE* gene expression (Figure 7A, group 2). Of note, however, none of the above differences were statistically significant.

Similarly, GBM patients with high CD8^+^ T cell abundance and high *CD69* gene expression (Figure 7B, group 4) showed better cumulative survival (>80 months), as compared to those patients with a low CD8^+^ T cells abundance and high *CD69* gene expression (Figure 7B, group 3). In contrast, those GBM patients with high CD8^+^ T cell abundance and low *CD69* expression (Figure 7B, group 2) exhibited reduced cumulative survival (approximately 20 months) compared to those patients with low CD8^+^ T cell abundance and low *CD69* expression (Figure 7B, group 1), who showed >40 months survival. However, none of the differences were statistically significant.

## 4. Discussion

GBM, the most common primary brain tumor, represents an extremely aggressive and highly infiltrating pathological entity which is still characterized by dramatically poor clinical outcomes in terms of survival [8,9]. Despite the progress made in recent years, the study of GBM is still particularly challenging from a diagnostic and therapeutic point of view. Where the new treatment strategies (e.g., immunotherapy) have brought significant benefits in terms of survival in subjects suffering from different types of cancer, they have failed or have shown unsatisfactory responses in GBM [8].

This phenomenon in GBM is probably due to the role of the TME in the maintenance and progression of the disease, and the lack or poor response to the main treatment protocols. For this reason, the objectives of our study were to characterize the tumor microenvironment of glioma with particular attention to TILs and, in this context, to identify and analyze the potential predictive and prognostic value and the possible therapeutic applications of TRM cells. In fact, these cells have recently come to light as important prognostic factors since several studies [13,14,16,23,24,25,26] have shown that their presence amongst TILs correlates with better survival in different types of cancer. 

Our results using flow cytometry, confocal microscopy, and bioinformatics highlight a statistically significant increase in the CD8^+^ TRM cell population in GBM tissues. 

In detail, the CD8^+^ (but not CD4^+^) CD103^+^ CD69^+^ TRM subset was significantly more abundant in GBM samples than in meningioma samples used herein as a control group. This finding was further supported by immunofluorescence/immunohistochemistry and gene expression analysis using available datasets, showing increased expression of *CD103* and *CD69*, two genes related to TRM CD8^+^ T lymphocytes. 

Hence these data, together with the positive correlation between *CD103* and *CD69* expression and CD8^+^ TILs abundance, indicate that TRM cells are an important component of the CD8^+^ TILs in GBM patients. Moreover, the significant negative correlations between *ITGAE* and *CD69* gene expression and molecular markers of GBM (*ITH2*, *ATRX,* and *TP53*) suggest a possible anti-tumoral role for CD8^+^ TRM T lymphocytes, similar to that found in different tumors [26,27,28,29].

Results of data mining transcriptomes and clinical files from a large cohort of GBM samples revealed that the cumulative survival was higher in GBM patients with a high number of tumor-infiltrating CD8^+^ T cells and *ITGAE* (CD103) and CD69 positive cells. Patients with high CD8^+^ T cell abundance and high ITGAE gene expression showed better cumulative survival than patients with a low CD8^+^ T cell number and high ITGAE gene expression. GBM patients with low CD8^+^ T cell abundance and low ITGAE expression had lower cumulative survival than patients with high CD8^+^ T cell abundance and low ITGAE gene expression. Again, patients with high CD8^+^ T cell abundance and high CD69 gene expression showed better cumulative survival than patients with a low CD8^+^ T cell number and high CD69 gene expression. GBM patients with high CD8^+^ T cell abundance and low CD69 expression exhibited significantly reduced cumulative survival compared to those with low CD8^+^ T cell abundance and low CD69 expression. Published data highlighted that the abundance of ITGAE expression and CD8, correlates with a poor prognosis in cutaneous squamous carcinoma and clear cell renal cell carcinoma [30,31], while in contrast, the same abundance correlates with a better prognosis in colorectal and ovarian cancer [32,33]. Taken together, these data highlight the presence of CD8^+^ TRM cells in GBM, leaving a question to be further investigated regarding their specific role in the context of this disease.

Altogether, these data, although not statistically significant, indicate the possibility that the maintenance of CD8^+^ TRM T lymphocytes within GBM TILs positively associate with better patient outcomes.

This work is a preliminary study and as such, it has some limitations. First, the small number of patients enrolled. Secondly, the lack of follow-up data and, consequently, the absence of data relating to the survival of the patients involved in the study. Lastly, the absence of functional studies of the TRM cells infiltrating GBM. Despite these limitations, we can state that CD8^+^ TRM cells are present among tumor-infiltrating lymphocytes and their frequency significantly increases in tumor tissue compared to control. In conclusion, our study clearly demonstrates that CD8^+^ TRM T lymphocytes are an important component of TILs in GBM patients.

## Figures and Tables

**Figure 1 biomedicines-10-02454-f001:**
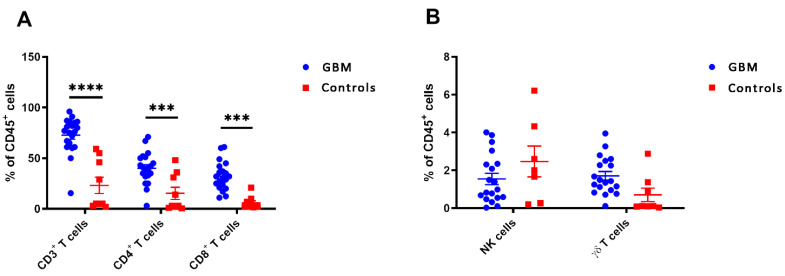
Cumulative analysis of the immune infiltrate of *n* = 20 GBM samples. The lymphocyte populations were identified by analyzing the expression of cell surface markers. (**A**) show the percentage of CD3^+^, CD4^+^ helper and CD8^+^ cytotoxic T lymphocytes, while (**B**) show the percentage of NK and γδ T cells in GBM and control samples. Data are reported as the average percentage value of the total CD45^+^ viable cells ± SE. Statistical analysis was performed by Multiple *t*-test Analysis, *** *p* < 0.001, **** *p* < 0.0001.

**Figure 2 biomedicines-10-02454-f002:**
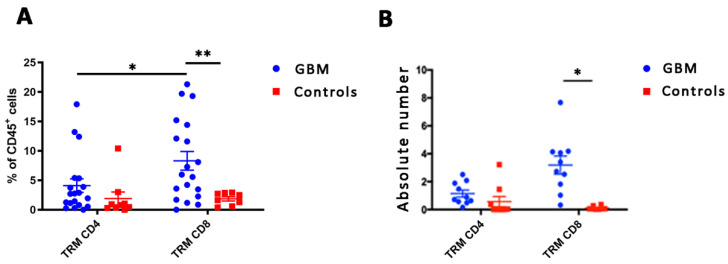
Cumulative analysis of the frequency (**A**) and absolute values (**B**) of TRM cells in 20 GBM and 9 meningioma samples. Values are expressed as mean percentage ± SE. Statistical analysis was performed by Multiple t-test Analysis, * *p* < 0.05, ** *p* < 0.01.

**Figure 3 biomedicines-10-02454-f003:**
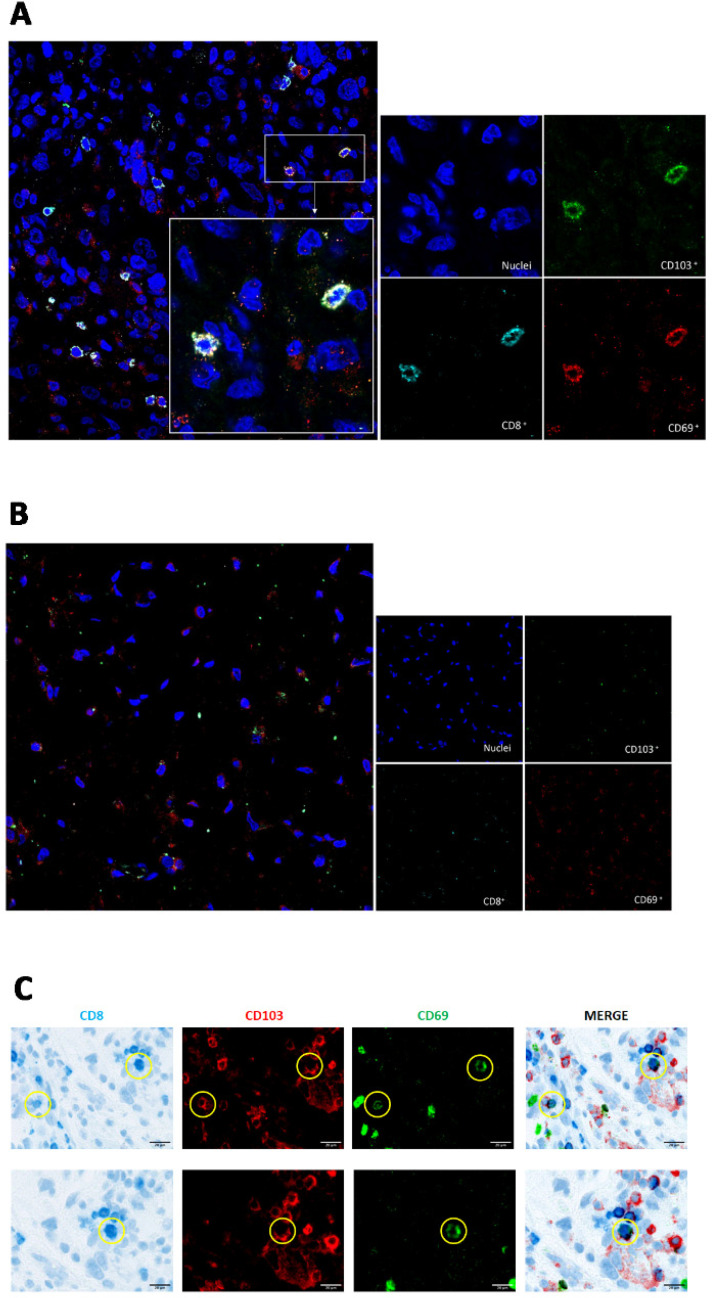
CD8^+^ TRM cells infiltrate tumor tissue in GBM patients. (**A**) confocal microscopy analysis from the CLADIBIOR laboratory shows the localization of the surface markers (CD103 green signal and CD69 red signal) of CD8^+^ TRM cells in a representative GBM sample, and (**B**) their absence in meningioma. Original magnifications, ×400, and ×630. (**C**) Two representative microphotographs from the Tumor Immunology Unit’s laboratory, of multiplexed immunostaining for CD8 (blue signal), CD103 (red signal), and CD69 (green signal) in human glioblastoma FFPE sample. Original magnifications, ×400 and ×630.

**Figure 4 biomedicines-10-02454-f004:**
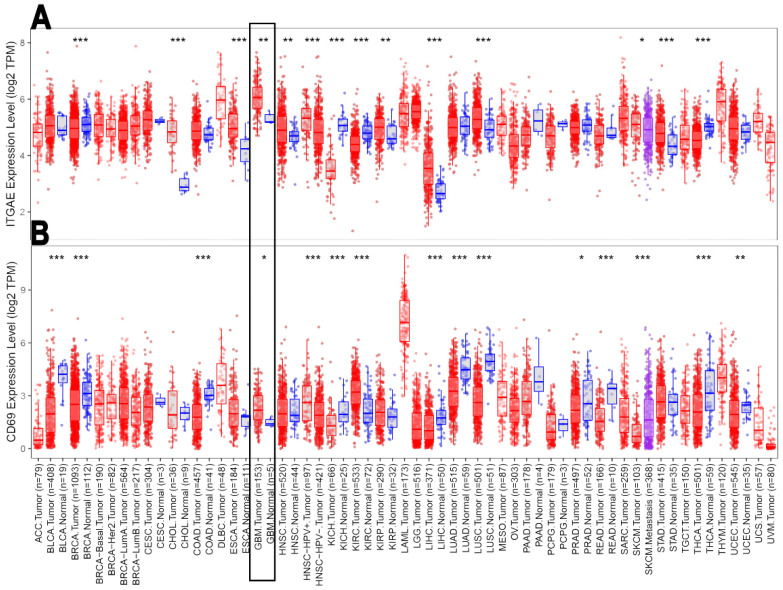
Wilcoxon test from on a cohort of *n* = 153 GBM patients and *n* = 5 controls. These two box plots show the differential gene expression between tumor tissue (red dots) and adjacent healthy tissues (blue dots) of (**A**) ITGAE, and (**B**) CD69. *: *p* < 0.05; **: *p* <0.01; ***: *p* <0.001.

**Figure 5 biomedicines-10-02454-f005:**
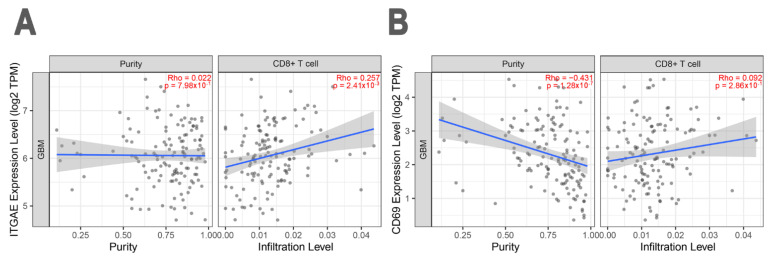
Spearman correlation test from Using EPIC on a cohort of *n* = 153 patients with GBM. Scatter plot showing the positive correlation between the abundance of CD8^+^ TILs and the expression of (**A**) *ITGAE* and (**B**) *CD69*, genes. In (**A**,**B**), the left panel indicates tumor purity, and the right panel indicates T cells infiltrate. Tumor purity Adjustment (Purity) and immune infiltration are used for negatively correlated.

**Figure 6 biomedicines-10-02454-f006:**
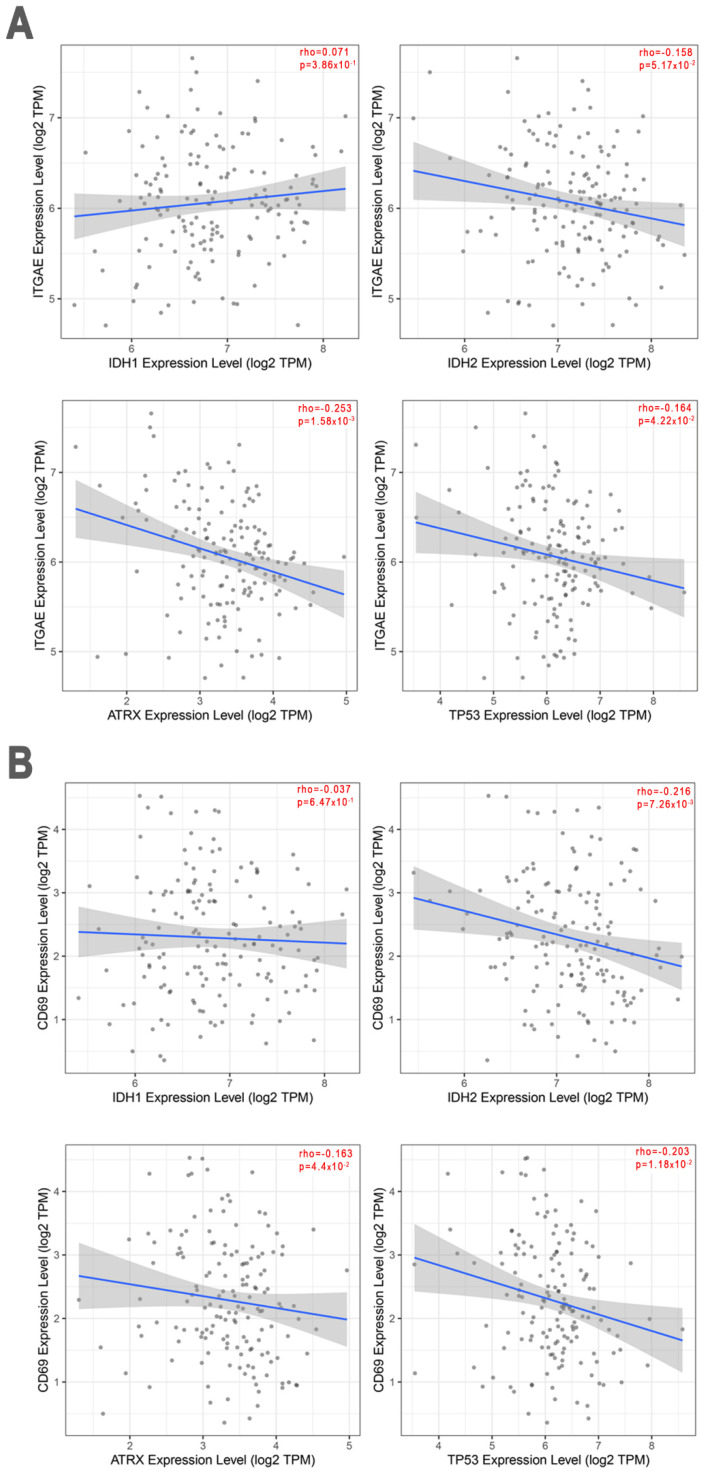
Spearman correlation test from EPIC on a cohort of *n* = 153 patients with GBM. Scatter plots show the correlations between *ITGAE* (**A**) or *CD69* (**B**) gene expression and the molecular markers of GBM.

**Figure 7 biomedicines-10-02454-f007:**
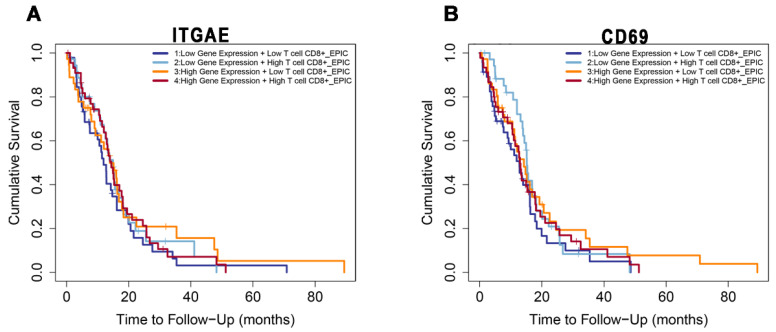
Kaplan-Meier curves from EPIC on a cohort of *n* = 153 patients with GBM and available follow-up. Shown is the correlation between *ITGAE* (**A**) and *CD69* (**B**) gene expression and the cumulative survival of patients, according to CD8^+^ TIL abundance. (**A**), group 4 vs. group 3: HR = 1.4, *p* = 0.201; group 2 vs. group 1: HR = 1.17, *p* = 0.534. (**B**) group 2 vs. group 1: HR = 1.76, *p* = 0.049; group 4 vs. group 3: HR = 1.05, *p* = 0.851.

## Data Availability

The flow cytometric, and the immunofluorescence data will be available upon reasonable request to the corresponding author. The data used for gene expression analysis are available at the National Cancer Institute—The Cancer Genome Atlas Program (TCGA) and the developed script for this study will be available upon reasonable request to the corresponding author.

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
