# Peer review of "The Abundance of Tumor-Infiltrating CD8+ Tissue Resident Memory T Lymphocytes Correlates with Patient Survival in Glioblastoma"

_biomedicines, 2022, doi:10.3390/biomedicines10102454_

Round 1

Reviewer 1 Report

The manuscript describes the separation and characterization of tumor environment biomarkers in clinical GBM samples. The authors use serial techniques, including flow cytometry, confocal microscopy, and histological analyses, to prove that the CD8+ tissue-resident memory T cells in GBM are significantly higher expressed than in the control samples. Furthermore, the authors find that the expression levels of CD8+ tissue-resident memory T lymphocytes within GBM tumor infiltrating lymphocytes positively correlate with patient survival. While the sample size is small, the study indicates that CD8+ TRM T lymphocytes probably can serve as an essential and reliable biomarker in the diagnosis and prognosis of GBM. Therefore, I recommend the manuscript for publication in Biomedicines after the authors address the following issues:

1.    The study included 20 GBM patients. Have the authors analyzed all patients' samples for confocal microscopy analysis?

2.       Please provide scale bars for the images in Figure 3.3.

3.       Do the authors expect clear cutoff values for the biomarkers to predict the survival rate after increasing the clinical sample size?

4.       Please pay attention to the reference and format, such as line 257, [19-21] [21-23].

5.       Please provide a full name for all abbreviations during the first appearance, such as ITGAE, FFPE, FACS.

Author Response

Reviewer 1

Open Review

The manuscript describes the separation and characterization of tumor environment biomarkers in clinical GBM samples. The authors use serial techniques, including flow cytometry, confocal microscopy, and histological analyses, to prove that the CD8+ tissue-resident memory T cells in GBM are significantly higher expressed than in the control samples. Furthermore, the authors find that the expression levels of CD8+ tissue-resident memory T lymphocytes within GBM tumor infiltrating lymphocytes positively correlate with patient survival. While the sample size is small, the study indicates that CD8+ TRM T lymphocytes probably can serve as an essential and reliable biomarker in the diagnosis and prognosis of GBM. Therefore, I recommend the manuscript for publication in Biomedicines after the authors address the following issues:

  1. The study included 20 GBM patients. Have the authors analyzed all patients' samples for confocal microscopy analysis?

We analyzed only 5 GBM and 2 meningioma among the samples, to assess the anatomical location of TRM cells.  This has now been clarified in the Results section.

  1. Please provide scale bars for the images in Figure 3.3.

We added the scale bars in the images of figure 3C.

  1. Do the authors expect clear cutoff values for the biomarkers to predict the survival rate after increasing the clinical sample size?

The point of this reviewer is well taken. Accordingly, we are working to increase the sample size to analyze by single-cell sequencing and flow cytometry TRM cells. Our purpose is to find out a clear cutoff in the expression of at least one of the studied biomarkers which correlate with survival among the patients.

  1. Please pay attention to the reference and format, such as line 257, [19-21] [21-23].

We thank the reviewer for the suggestion. We have now fixed the reference format at line 257.

  1. Please provide a full name for all abbreviations during the first appearance, such as ITGAE, FFPE, FACS.

We thank the reviewer for the suggestion. We have now provided the full name of all abbreviations, including ITGAE, FFPE and FACS, in the manuscript.

Reviewer 2 Report

This manuscript reports a series of studies investigating glioblastoma multiforme tumors and a sub-population of tumor-infiltrating lymphocytes called TRM. The authors combine analysis of tissue samples from their own medical system and in silico analysis of tumor samples present in the TCGA database. They find that GBM tumors are enriched in TRM. The manuscript is well written, and the subject is an interesting one. I have some concerns about the conclusions drawn about correlation between TRM presence and survival in GBM patients, and have a few other comments.

- For data shown in Figure 3, was there any attempt to quantify the number of CD103/CD69 co-expressing TRM cells in the samples, such as by cell counting? Were there not any co-expressing cells present in meningioma samples?

- Page 8, line 290-291: The correlation statistics mentioned in the text do not match those in the graphs in Figure 5. These should be double-checked to make sure they agree.  

- The main issue that I see with the manuscript as written is that the data do not really support (or refute) the major conclusion that CD8+ TRM “positively associate with better patients outcomes.” This appears to be taken largely from Figure 7 data. However, the only statistically significant comparison reported is a difference between low CD69 expression/low CD8+ TIL numbers and low CD69 expression/high CD8+ TIL numbers. Other differences cited (e.g. those in Figure 7A) are not statistically significant, so should not be used to support a conclusion. I would expect a survival effect of these cells to show up more strongly in the high/high groups; reasons for this should probably be explored in the discussion.  

- Also related to Figure 7, it is not clear to me what criteria signify high or low CD8+ numbers or high or low gene expression. Were these data sets dichotomized? If so, what were the criteria for differentiating the groups? This could be as simple as just using a median split between the groups, but it should be disclosed how it was done.

Minor notes:

- There are a number of acronyms and abbreviations that are used without defining. These should checked and defined.

- The authors have placed a short discussion of limitations in the conclusion section. This is fine if desired, but they could also consider moving this to the discussion section.

- Page 4, line 152-153: Supplemental figure 1 is cited, but does not appear to have been included.

- For TCGA analyses, the database website should be included as a citation.

- Page 6, lines 243-244: It does not appear that abundance of CD4+ TRM cells was compared to CD8+ TRM cells statistically. Thus, it is not correct to say that one is more abundant than the other.

- Figure 6 is hard to read because the graphs and their text is very small. I suggest changing the orientation of the graphs so the figure can be made larger. 

Author Response

Reviewer 2

Comments and Suggestions for Authors

This manuscript reports a series of studies investigating glioblastoma multiforme tumors and a sub-population of tumor-infiltrating lymphocytes called TRM. The authors combine analysis of tissue samples from their own medical system and in silico analysis of tumor samples present in the TCGA database. They find that GBM tumors are enriched in TRM. The manuscript is well written, and the subject is an interesting one. I have some concerns about the conclusions drawn about correlation between TRM presence and survival in GBM patients, and have a few other comments.

- For data shown in Figure 3, was there any attempt to quantify the number of CD103/CD69 co-expressing TRM cells in the samples, such as by cell counting? Were there not any co-expressing cells present in meningioma samples?

We did not assess absolute values of double positive CD69/CD103 cells either in the GBM or in meningioma control slide in figure 3A. The reason is that we only analyzed 5 GBM and 2 meningioma among the samples, to assess the anatomical location of TRM cells.  This has now been clarified in the Results section.

- Page 8, line 290-291: The correlation statistics mentioned in the text do not match those in the graphs in  Figure 5. These should be double-checked to make sure they agree. 

We apologize for this mistake. We have now fixed the error and rephrased the period.

- The main issue that I see with the manuscript as written is that the data do not really support (or refute) the major conclusion that CD8+ TRM “positively associate with better patients’ outcomes.” This appears to be taken largely from Figure 7 data. However, the only statistically significant comparison reported is a difference between low CD69 expression/low CD8+ TIL numbers and low CD69 expression/high CD8+ TIL numbers. Other differences cited (e.g. those in Figure 7A) are not statistically significant, so should not be used to support a conclusion. I would expect a survival effect of these cells to show up more strongly in the high/high groups; reasons for this should probably be explored in the discussion. 

We fully agree with the point of the reviewer and we have changed the conclusion regarding the significance of the data in figure 7.

- Also related to Figure 7, it is not clear to me what criteria signify high or low CD8+ numbers or high or low gene expression. Were these data sets dichotomized? If so, what were the criteria for differentiating the groups? This could be as simple as just using a median split between the groups, but it should be disclosed how it was done.

We appreciate the reviewer’s accuracy on this matter.  Data in Figure 7 were not dichotomized: We used an algorithm based on PCA and Dimensionality reduction methods, so a median split cannot be applied.  Accepting the reviewer’s suggestion, we have described this issue in the Methods section.

Minor notes:

- There are a number of acronyms and abbreviations that are used without defining. These should checked and defined.

We checked and defined all the acronyms and abbreviations.  We thank you the reviewer.

- The authors have placed a short discussion of limitations in the conclusion section. This is fine if desired, but they could also consider moving this to the discussion section.

We thank the reviewer for the suggestion.  Accordingly, we have moved the conclusion into the discussion section.

- Page 4, line 152-153: Supplemental figure 1 is cited, but does not appear to have been included.

We have submitted Supplemental Figure 1 as a separate file. We have now included Supplemental figure 1 in the text, at the end of the manuscript.

- For TCGA analyses, the database website should be included as a citation.

We appreciate the suggestion, and the database website has been cited accordingly.

- Page 6, lines 243-244: It does not appear that abundance of CD4+ TRM cells was compared to CD8+ TRM cells statistically. Thus, it is not correct to say that one is more abundant than the other.

We apologize for the error. Actually, the difference between CD4+ and CD8+ T cells frequencies in GBM is statistically significant. We have now fixed this error.

- Figure 6 is hard to read because the graphs and their text is very small. I suggest changing the orientation of the graphs so the figure can be made larger.

We changed the orientation of Figure 6 according to the reviewer’s suggestion. We hope now it is easier for the reader to follow.

Round 2

Reviewer 2 Report

The authors have addressed my concerns with the manuscript.

I did note that in line 310, the p value and R value appear to have been reversed (based on what is shown on the graph in Fig. 5). This is clearly just a typo. 

Also, in the legend for Supplementary Figure 1, I believe the authors meant to write "dot plots" rather than "do plots." 

Author Response

I did note that in line 310, the p value and R value appear to have been reversed (based on what is shown on the graph in Fig. 5). This is clearly just a typo. 

We have reversed the two values, we thank  the referee.

Also, in the legend for Supplementary Figure 1, I believe the authors meant to write "dot plots" rather than "do plots." 

We have fixed the error in the legend of supplemental Figure 1.